# Immune Heterogeneity Between Primary Tumors and Corresponding Metastatic Lesions and Response to Platinum Therapy in Primary Ovarian Cancer

**DOI:** 10.3390/cancers11091250

**Published:** 2019-08-26

**Authors:** Katharina Dötzer, Friederike Schlüter, Markus Bo Schoenberg, Alexandr V. Bazhin, Franz Edler von Koch, Andreas Schnelzer, Sabine Anthuber, Dieter Grab, Bastian Czogalla, Alexander Burges, Jens Werner, Sven Mahner, Barbara Mayer

**Affiliations:** 1Department of General, Visceral and Transplant Surgery, Ludwig-Maximilians-University Munich, Marchioninistraße 15, 81377 Munich, Germany; 2German Cancer Consortium (DKTK), Partner Site Munich, Pettenkoferstraße 8a, 80336 Munich, Germany; 3Department of Obstetrics and Gynecology, Klinikum Dritter Orden, Menzinger Straße 44, 80638 Munich, Germany; 4Department of Obstetrics and Gynecology, Klinikum rechts der Isar, Technical University Munich, Ismaninger Straße 22, 81675 Munich, Germany; 5Department of Obstetrics and Gynecology, Clinic Starnberg, Oßwaldstraße 1, 82319 Starnberg, Germany; 6Department of Obstetrics and Gynecology, Clinic Harlaching, Sanatoriumsplatz 2, 81545 Munich, Germany; 7Department of Obstetrics and Gynecology, University Hospital, Ludwig-Maximilians-University Munich, Marchioninistraße 15, 81377 Munich, Germany

**Keywords:** ovarian cancer, metastatic lesions, tumor microenvironment, TILs, immune heterogeneity, platinum-sensitivity, immune checkpoints

## Abstract

CD3^+^ and CD8^+^ lymphocytes are well known prognostic markers in primary ovarian cancer. In contrast, the predictive value of the immune infiltrate concerning treatment response and the involvement of immune heterogeneity between primary and metastatic lesions are poorly understood. In this study, the immune infiltrate of 49 primary tumors and 38 corresponding lesions in the omentum (*n* = 23) and the peritoneum (*n* = 15) was immunohistochemically analyzed and correlated with clinicopathological factors and platinum-sensitivity. Immune heterogeneity was observed between paired primary and metastatic lesions for all immune cell phenotypes. The stromal immune infiltrate was higher in the omental lesions than in the primary tumors, which was reflected by CD45 (*p* = 0.007), CD3 (*p* = 0.005), CD8 (*p* = 0.012), and PD-1 (programmed cell-death protein 1) (*p* = 0.013). A higher stromal infiltrate of both CD45^+^ and CD3^+^ cells in the omental lesions was associated with the detection of lymph node metastasis (CD45, *p* = 0.018; CD3, *p* = 0.037). Platinum-sensitive ovarian cancers revealed a higher intratumoral CD8^+^ infiltrate in the peritoneal lesions compared to the primary tumors (*p* = 0.045). In contrast, higher counts of stromal PD-1^+^ cells in the peritoneal lesions have been associated with reduced platinum-sensitivity (*p* = 0.045). Immune heterogeneity was associated with platinum response and might represent a selection marker for personalized therapy.

## 1. Introduction

High cell counts of various immune cell markers in ovarian cancer have been identified as positive prognostic factors. An especially high intratumoral infiltrate of CD3^+^ and CD8^+^ cells is associated with increased progression free survival (PFS) and overall survival (OS). This has been investigated in several studies [1,2]. Whereas the prognostic value of immune infiltrate in ovarian cancer has been evaluated in detail, there have been less studies investigating the predictive value for treatment response. More than two-thirds of the patients experience relapse within the first three years, despite optimal surgery and adjuvant chemotherapy with carboplatin and paclitaxel [3]. 

Incorporation of biological drugs may improve PFS [4,5]. Tumor biology has an increasing impact in the treatment of recurrent ovarian cancer, extending the classification according to time to relapse after chemotherapy: platinum-resistant (<6 months), partially platinum-sensitive (6–12 months), and platinum-sensitive (>12 months) [6,7]. This clinical classification has been used when choosing second-line chemotherapy [8]. Further stratification factors are required for specifying first-line therapy [9]. Evaluation of the impact of the immune infiltrate on chemosensitivity might help to select the most appropriate patients for treatment with chemotherapy. In addition, the immune infiltrate might be a predictive marker for immunotherapy. 

As a result of the discovery of lymphocytes as an important factor in the antitumoral defence of ovarian cancer, checkpoint inhibitors have rapidly emerged in past years in new therapeutic approaches [10,11,12,13]. Recently, various studies about treatment with checkpoint inhibitors in combination with platinum-based chemotherapy have started, investigating both primary ovarian cancer (NCT02718417, NCT02520154, NCT02766582) and the recurrent situation (NCT02891824) [14]. As none of these trials considers specific biomarkers to preselect appropriate patients, there is an obvious need to investigate the immune infiltrate in primary tumors. Because most ovarian cancer patients are diagnosed in an advanced stage [15], metastatic lesions in the peritoneum and omentum need to be analyzed in addition. While genomic, transcriptomic, and proteomic intratumoral heterogeneity are well published in ovarian cancer [16,17,18,19], immune heterogeneity has not been systematically investigated. 

In the present study, various immune cell phenotypes were analyzed in the primary tumor and compared with metastatic lesions in the peritoneum and the omentum. In detail, the density of CD45^+^, CD3^+^, CD8^+^, PD-1^+^ (programmed cell-death protein 1), and PD-L1^+^ (programmed cell-death ligand 1) cells was evaluated in the stromal and intratumoral areas of the different lesions and correlated with a number of clinicopathological factors and the platinum-sensitivity. Heterogenous distribution of the immune infiltrate might impact treatment management. 

## 2. Results

### 2.1. Patient Characteristics

Clinicopathological characteristics are shown in Table 1. Most patients presented with high-grade, serous ovarian carcinoma and have been diagnosed in an advanced FIGO (International Federation of Gynaecology and Obstetrics) stage, with the presence of ascites and lymphatic vessel invasion. Seventy-one percent of all patients had a complete surgical resection of all macroscopic visible tumor. In accordance with the advanced tumor stage, 82% of the patients received a carboplatin–paclitaxel-based chemotherapy. The median PFS was 19 months (range 9–42). Concerning time to relapse after chemotherapy, 14 patients were defined as reduced platinum-sensitive (tumor relapsed ≤12 months after chemotherapy) and 28 patients as full platinum-sensitive (tumor relapsed >12 months after chemotherapy).

The presence of distant metastases at time of diagnosis was significantly associated with reduced platinum-sensitivity (*p* = 0.015, Appendix A). The presence of ascites before surgery (*p* = 0.083) and macroscopic residual tumor after surgery (*p* = 0.067) showed a trend to reduced platinum-sensitivity. 

In accordance with these results, the presence of metastases (*p_log-rank_* = 0.031, *p_Breslow_* = 0.035), macroscopic residual tumor after surgery (*p_log-rank_* = 0.01, *p_Breslow_* = 0.005), and vascular invasion (*p_log-rank_* = 0.006, *p_Breslow_* = 0.03) correlated significantly with shorter PFS (Appendix A).

### 2.2. Immune Infiltrate in Primary Tumor

All immune cell phenotypes were detected in the stromal area of the primary tumor in a higher fraction compared to the intratumoral area. This finding was independent from the method of evaluation. The highest density was observed for CD45^+^ cells, followed by CD3^+^ cells, CD8^+^ cells, and PD-1^+^ cells (Table 2). 

Patients who showed characteristics of tumor progression, i.e., vascular invasion, the detection of distant metastasis and the presence of ascites, at the time of diagnosis showed a strong leukocyte infiltrate. A high density (rating ≥3) of CD45^+^ leukocytes was frequently observed in the stromal area of primary tumors with distant metastasis (cM1, 93%, Appendix A). Contrary, stromal CD45^+^ cells were less frequently found in cancers without distant metastasis (cM0, 63%, *p* = 0.042). All patients with a strong (rating ≥2) CD45^+^ intratumoral infiltrate in primary tumor were suffering from ascites (*p* = 0.006). Vascular invasion significantly correlated with a high (>73 counts/mm^2^) density of PD-1^+^ cells in the stromal area of the primary tumor (*p* = 0.013). PD-L1 positivity was found more often in primary tumors with distant metastasis (86%) compared to cancers without distant metastasis (51%, *p* = 0.049). A high intratumoral density (>88 counts/mm^2^) of CD8^+^ cells was predominantly observed in older patients (>62 years, 78%, *p* = 0.037).

Primary tumors with a high (>201 counts/mm^2^) intratumoral CD3^+^ cell density showed a trend to full platinum-sensitivity (*p* = 0.057, Table 3).

### 2.3. Immune Infiltrate in Metastatic Lesions

Direct comparison of primary tumors and omental lesions revealed a higher rating for stromal CD45^+^ cells in the omental lesion in more than half of the patients (52%, *p* = 0.007, Appendix A). Furthermore, in 16 cases (70%), the omental lesion showed a higher density of stromal CD3^+^ and CD8^+^ cells compared to the primary tumor (*p* = 0.005 and *p* = 0.012, Figure 1). Consequently, the mean count in omental lesions of stromal CD3^+^ and CD8^+^ cells was nearly two times higher. In addition, the majority of omental lesions (65%) revealed a higher infiltrate of stromal PD-1^+^ cells (*p* = 0.013). There was no significant difference in intratumoral counts comparing primary tumors and omental lesions.

In the stromal area of primary tumors and corresponding peritoneal lesions, no significant differences in the immune infiltrate could be found. In contrast, in ten cases (67%), the counts of intratumoral PD-1^+^ cells were higher in the primary tumor compared to the corresponding peritoneal lesion (*p* = 0.054, Figure 2). Conversely, only one case (7%) showed a higher expression of PD-L1 in the primary tumor, while it was equal or lower in most patients (93%, *p* = 0.074).

### 2.4. Associations of Immune Heterogeneity

Patients with lymph node metastases revealed a strong stromal immune infiltrate in the omental lesion. Lymph node metastases have been found at the time of diagnosis in all patients with a higher immune infiltrate of stromal CD45^+^ cells in the omental lesion compared to the primary tumor (*n* = 11, *p* = 0.018, Appendix A). Most tumors with lymph node metastases (87%) revealed more stromal CD3^+^ cells in the omental lesion than in the primary tumor (*p* = 0.037). No significant correlations have been found for peritoneal lesions (Appendix A). Interestingly, immune heterogeneity between peritoneal lesions and primary tumors could be identified as a predictive marker. All patients with reduced platinum-sensitivity showed a stronger stromal infiltrate of PD-1^+^ cells in the peritoneal lesion than in the primary tumor (*p* = 0.045, Table 3). In contrast, higher counts of intratumoral CD8^+^ cells in the peritoneal lesions were associated with full platinum-sensitivity (*p* = 0.045).

## 3. Discussion

The urgent need for sustainable therapeutic strategies in ovarian cancer has triggered a comprehensive search for predictive biological markers. A number of new candidates have been identified in primary tumors [20]. In contrast, corresponding metastatic lesions are rarely analyzed although it is well known that the microenvironment has a profound impact on tumor cell biology and therapeutic response [21,22]. For the first time, in the present study, a systematic side-by-side comparison of the quantitative immune infiltrate in the primary tumor and metastatic lesions located in the omentum and peritoneum was performed and correlated with clinicopathological factors. In primary tumors, the presence of a strong CD45^+^ infiltrate in both the stromal and intratumoral compartments correlated with parameters of advanced disease at time of initial diagnosis. In addition, strong PD-L1 expression in primary tumors correlated with metastatic disease. The presence of a strong stromal PD-1^+^ infiltrate was frequently observed in primary tumors characterized by vascular invasion, suggesting an immunosuppressive role in advanced primary ovarian cancer. Indeed, a high expression of PD-1 and PD-L1 was reported to be associated with poor prognosis [23,24,25]. An increased intratumoral CD3^+^ infiltrate in chemo-naive primary tumors was associated with response to platinum-based chemotherapy. This supports the 2011 study from Bösmüller et al., which indicated intratumoral lymphocyte density superior to ERCC-1 (excision repair cross-complementation group 1) expression to predict platinum response [26]. A number of different mechanisms is known to be involved in an increased antitumoral activity of lymphocytes induced by platinum-based chemotherapy [27,28]. Consequently, a pre-therapeutic increased number of CD3^+^ cells might be associated with a higher efficacy. Methods to enhance this effect could be the application of dose-dense chemotherapy [29]. Stratification of primary tumors according to a strong CD3^+^ infiltrate and a high PD-1/PD-L1 expression might identify ovarian cancer patients who are responsive to a combination of platinum-based chemotherapy with checkpoint-inhibitors [30,31]. 

In more than 70% of the patients, disease has already spread in peritoneal and omental tissue at the time of diagnosis, which is associated with an unfavorable prognosis [15,32]. A higher fraction of CD45^+^, CD3^+^, CD8^+^, and PD-1^+^ cells was observed in the stroma of omental lesions compared to the corresponding primary tumors. Higher counts of stromal CD45^+^ and CD3^+^ cells in the omental lesions were associated with the presence of lymph node metastases. Indeed, aggregates of leukocytes named milky spots are characteristic for the omental tissue and were found to promote immune suppression and, in consequence, facilitate tumor cell implantation [33,34]. They are discussed as a potential target for immunotherapy [35]. Combining strategies that enforce intratumoral recruitment of CD8^+^ cells might, therefore, increase immunotherapeutic effects [36,37,38].

Similar to omental lesions, comparison between peritoneal lesions and their corresponding primary tumors identified changes in the immune cell infiltrate, i.e., less intratumoral PD-1^+^ cells and a higher PD-L1 expression in the peritoneal metastases. Moreover, peritoneal lesions were characterized by a higher infiltration of intratumoral CD8^+^ cells, which correlated with the response to the platinum-based first-line treatment. For example, in patient 5082, a high density (30/mm^2^) of intratumoral CD8^+^ cells was found in the peritoneal lesion, while there was a low intratumoral CD8^+^ cells density (7/mm^2^) in the corresponding primary tumor. In this individual patient platinum-based chemotherapy was successful, resulting in a relapse-free time of 26 months after chemotherapy. Conversely, in the peritoneal lesion of patient 5072, a low number (2/mm^2^) of intratumoral CD8^+^ cells was found, while a high infiltrate (46/mm^2^) was found in the primary tumor. This patient suffered from a relapse within six months after platinum-based chemotherapy. Thus, for patients with a low peritoneal infiltrate, other therapeutic options than platinum-based chemotherapy should be considered right from the start. Treatment options might be non-platinum chemotherapeutic drugs or targeted therapies.

Conversely to the findings concerning intratumoral CD8^+^ cells, a higher density of stromal PD-1^+^ cells in the peritoneal lesions compared to the primary tumors was associated with a reduced platinum-sensitivity, which might be explained by impairment of these cells [23,24]. This further supports the close correlation between immune heterogeneity and platinum response. Comparing the immune contexture, defined as density, composition, and functionality of the immune infiltrate [39], in omental, peritoneal, and primary lesions, profound differences were observed. These changes might go along with dynamic variations in the tumor microenvironment at the different locations. In fact, the peritoneal cavity is characterized by complex multicellular interactions regulated by a network of multiple soluble factors [22,40,41]. 

Although the cohort of the present study can be described as representative and is comparable to cohorts of other clinical trials [42,43,44], the main limitation is the small number of patients. In particular, the number of samples of omental and peritoneal tissue was low. To confirm the results of the current pilot study, further analysis of immune heterogeneity should be pursued in a larger cohort. This study indicates that immune heterogeneity represents an important characteristic of ovarian cancer. Location-dependent changes in the immune cell infiltrate were identified as potential predictive markers for standard chemotherapy and immunotherapy. Thus, the systematic analysis of the immune contexture in both primary tumors and the corresponding metastatic lesions could have a relevant impact on treatment planning for individual cancer patients. 

## 4. Patients and Methods

### 4.1. Study Population

Forty-nine patients diagnosed with a primary, chemo-naive ovarian, fallopian tube, or peritoneal cancer from the SpheroID-Study were included. Patients suffering from another neoplasia within the last five years were excluded. Informed consent was obtained from all patients. The study was approved by the institutional review board of the Ludwig-Maximilians-Universität Munich, Munich, Germany (No.278/04). Patients were recruited between September 2012 and January 2015 in five ovarian cancer centers, namely University Hospital, LMU Munich (*n* = 16), Klinikum Dritter Orden (*n* = 16), Klinikum rechts der Isar, Technical University Munich (*n* = 7), Clinic Harlaching (*n* = 5), and Clinic Starnberg (*n* = 5). Standardized surgical resection and pathological analysis was conducted by the recruiting clinic. In 33 cases, in addition to primary tumor samples, corresponding omental or peritoneal lesions were resected. A detailed description of the tumor samples analyzed is given in Appendix A. Patient-, tumor-, and treatment-related data for correlations was given in the routine reports and delivered in a pseudonymized form. Subsequently, for each clinicopathological parameter, patients were divided according to clinical relevance into two groups. For the parameter ‘age’, the mean value was chosen as cutoff. Analysis of platinum-sensitivity was performed after completion of chemotherapy [7,45]. Seven patients with no chemotherapy or a reduced number of treatment cycles (≤2) had to be excluded. In the present cohort, patients with platinum-resistant disease (*n* = 2) and patients with partially platinum-resistant disease (*n* = 12) were grouped and defined as reduced platinum-sensitive. This group was compared to the full platinum-sensitive group (*n* = 28) [6,7]. PFS was defined as the time from surgical treatment to the time of relapse or progression. As the median follow-up time was 26 months, OS has not been analyzed. Data from patients who did not die and had no relapse or progression was censored at the date of their last visit. 

### 4.2. Immunohistochemistry

After surgical removal, tumor samples were snap frozen in liquid nitrogen. Serial cryosections (5 µm) were performed, fixed in acetone, and stained immunohistochemically using the avidin-biotin-peroxidase complex method [46]. Briefly, nonspecific binding Fc-regions were saturated with 10%-AB-Serum in phosphate-buffered-saline (Bio-Rad, Hercules, CA, USA, 805135) for 20 min. Endogenous biotin was blocked by the Avidin-/Biotin-Blocking-Kit (Vector Laboratories, Burlingame, CA, USA, SP-2001) according to the manufacturer. Primary antibodies were incubated for 60 min at room temperature, namely clone 2B11 + PD7/26 directed against CD45 (Dako, Santa Clara, CA, USA, M0701, working concentration [wc] 4.5 μg/mL), clone UCHT1 directed against CD3 (BD Biosciences, San Jose, CA, USA, 550368, wc 1.25 μg/mL), clone C8/144B directed against CD8 (Dako, M7103, wc 3 μg/mL), clone MIH4 directed against PD-1 (affymetrix eBiosciences, San Diego, CA, USA, 14-9969, wc 10 μg/mL) and clone MIH1 directed against PD-L1 (affymetrix eBiosciences, 14-5983, wc 10 μg/mL). The antibody Ber-EP4 directed against EpCAM (Dako, M0804, wc 2.5 μg/mL) was used to define tumor areas. The antibody MOPC21 (Sigma-Aldrich, St. Louis, MO, USA, M9269) was used as Immunoglobulin G1 isotype control in the relevant wc. Antibodies directed against CD45, CD3, and EpCAM were incubated with the secondary biotinylated antibody (Jackson ImmunoResearch, Cambridgeshire, U.K., 315-065-048, wc 0.75 μg/mL) for 30 min and, subsequently, with peroxidase-conjugated streptavidin (Jackson ImmunoResearch, 016-030-084, wc 1 μg/mL) for 30 min. Antibodies directed against CD8, PD-1, and PD-L1 were incubated with the ZytoChem Plus HRP-Kit (Zytomed Systems, Berlin, Germany, HRP060), as recommended by the manufacturer. Antigen–antibody reaction was visualized by 3-amin-9-ethylcarabazol (AEC, Sigma-Aldrich, A5754)-peroxide-solution for eight min in darkness. Sections were counterstained with Mayers hemalum solution (Merck, Darmstadt, Germany, 109249).

### 4.3. Semiquantitative Analysis of the CD45^+^ Infiltrate

CD45^+^ cells were detected as single cells and often in cell clusters of different size. This staining pattern could not be dissected precisely by the software ImageJ (Version 1.51h, National Institutes of Health, Bethesda, MD, U.S). Therefore, the CD45 staining was semiquantitatively rated in two different spatial areas (Appendix A). In the intratumoral area, CD45^+^ cells were found in direct contact with the cancer cells. In the stromal area adjacent to the tumor compartment, CD45^+^ cells did not touch the tumor cells. Analysis was performed with the microscope (BX41, Olympus Corporation, Tokio, Japan).

### 4.4. Quantitative Analysis of Immune Cells

Quantitative analysis was performed for CD3, CD8, and PD-1. For each tissue section and each immune cell antigen, three separate regions characterized by an enriched infiltrate were selected, separately for the stromal and the intratumoral area (defined previously and demonstrated in Figure 3). CD8^+^ cells were counted in the same region as CD3^+^ cells. For PD-1, known to be expressed on different cell types [47], enriched regions were evaluated independently from the CD3/CD8 area. A picture of each enriched region was captured (×200 magnification) using Zen 2.0 lite software (Carl Zeiss Inc., Oberkochen, Germany) with an AxioCam MRc5 camera (Carl Zeiss Inc., Oberkochen, Germany). 

Counting was done according to the QTiS algorithm by Miksch et al. [48,49,50,51] with the ImageJ software (Version 1.51h, National Institutes of Health, Bethesda, MD, USA). Briefly, after substraction of background, color deconvolution was performed, selecting example regions for the following spectra: background, hematoxylin, and AEC. In the AEC-spectrum an automatic threshold was applied to get a binary image. Watershed method was used to separate cell clusters. Cells were counted with the ‘Analyze-Particle-Tool’, considering the values of ‘700-infitiy’ for size and ‘0.2–1.0’ for circularity. The ‘ROI-Manager’ was used to distinguish between intratumoral and stromal area. For each tissue section, an average count of the three images that were taken was calculated. Cell counts were normalized to an average number of cells/mm^2^.

The reliability of the semiquantitative rating system for CD45^+^ cells was checked by correlation of ratings and counts of CD3^+^ cells intratumoral and stromal. Spearman correlation coefficient was 0.803 (*p* = 0.000).

### 4.5. Semiquantitative Analysis of PD-L1

The fraction of all PD-L1^+^ cells, including both cancer and immune cells in the intratumoral area, was estimated. PD-L1 positivity was defined as ≥1%. This approach has already been used in former clinical trials [52,53].

### 4.6. Statistical Analysis

Clinicopathological factors were grouped by clinical relevance. For the immune cell phenotypes CD3, CD8, and PD-1 identified in the respective spatial area the mean cell count was defined as cutoff to distinguish low infiltrates from high infiltrates. The cutoff for CD45^+^ cells was chosen considering balanced group sizes (intratumoral: ≤1; >1; stromal: ≤2; >2). For comparison between corresponding location and primary tumor, the ratio between the cell counts was calculated for each immune cell phenotype in both spatial areas. To form two groups—‘corresponding location lower/equal than primary tumor’ and ‘corresponding location higher than primary tumor’—≤1; >1 was chosen as the cutoff. The defined cutoffs were correlated with various clinicopathological data and platinum-sensitivity using the Fisher’s exact two-tailed test. Cumulative survival probabilities were calculated by the Kaplan–Meier method. Log-rank and Breslow tests were used to compare clinicopathological factors regarding PFS. Immune cell phenotypes between the primary tumor and different corresponding locations were compared by Wilcoxon signed-rank test. *p*-values < 0.050 were considered to be statistically significant; *p*-values < 0.100 have been reported. All statistical analyses were performed using SPSS Statistics (Version 23.0, IBM, Armonk, NY, USA).

## 5. Conclusions

Immunohistochemical comparison in ovarian cancer between primary tumors and the corresponding omental and peritoneal lesions demonstrated profound differences in the expression pattern of CD45, CD3, CD8, PD-1 and PD-L1. This immune heterogeneity made an impact on platinum-sensitivity. Variations in the immunologic tumor microenvironment might influence treatment selection for the individual ovarian cancer patient. 

## Figures and Tables

**Figure 1 cancers-11-01250-f001:**
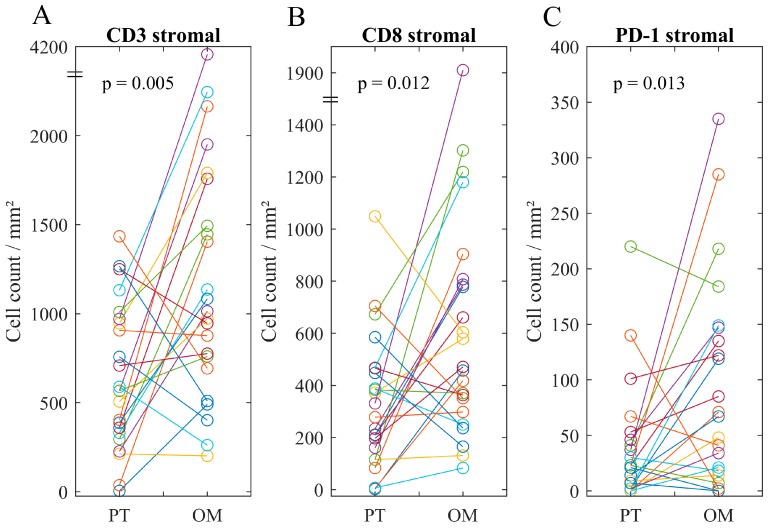
Scatter plots comparing immune cell phenotypes between primary tumor and the corresponding omental lesion. Counts of (**A**) CD3^+^, (**B**) CD8^+^, and (**C**) PD-1^+^ (programmed cell-death protein 1) stromal cells. Counts of CD3^+^, CD8^+^, and PD-1^+^ cells have been significantly higher in the omental lesions. *p*-value calculated by Wilcoxon signed-rank test.

**Figure 2 cancers-11-01250-f002:**
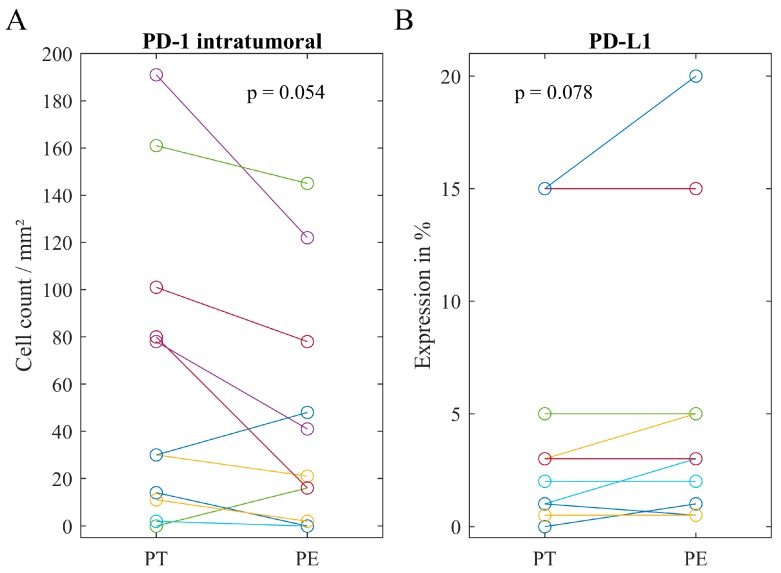
Scatter plots comparing immune cell phenotypes between primary tumor and the corresponding peritoneal lesion. (**A**) Counts of intratumoral PD-1^+^ (programmed cell-death protein 1) cells have been in tendencies higher in primary tumor. (**B**) PD-L1 (programmed cell-death ligand 1) expression has been slightly higher in peritoneum. *p*-value calculated by Wilcoxon signed-rank test. Cases with the value ‘0’ in primary tumor and peritoneum (A) n = 3 (B) n = 2 have been excluded from graph, but not from calculation.

**Figure 3 cancers-11-01250-f003:**
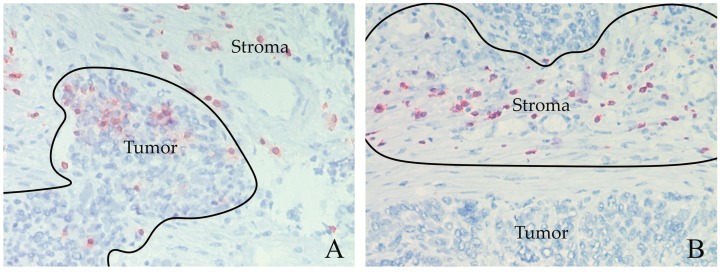
Immunohistochemical analysis demonstrating intratumoral (**A**) and stromal (**B**) CD3^+^ cells. The black borders mark the enriched regions, in which CD3^+^ cells were counted. Tumor and stroma areas are indicated. Magnification 200×.

**Table 1 cancers-11-01250-t001:** Patient characteristics.

		n or Value	%
**Age**	mean/median	62/66 years	
	range	24–83 years	
**FIGO Stage**	I/II	0	0.0
	III	35	71.4
	IV	14	28.6
**pT**	pT2	5	10.2
	pT3	44	89.8
**pN**	pN0	6	12.2
	pN1	32	65.3
	Nx	11	22.4
**cM**	cM0	35	71.4
	cM1	14	28.6
**Primary Tumor Site**	Ovarian	39	79.6
	Fallopian Tube	7	14.3
	Peritoneal	3	6.1
**Histological Subtype**	Serous	44	89.8
	Other	5	10.2
**Grading**	G1/G2	1	4.0
	G3	47	95.9
**Ascites**	yes	41	83.7
	no	8	16.3
**Macroscopic Residual Tumor after Surgery**	None	35	71.4
	<1 cm	8	16.3
	>1 cm	6	12.2
**Lymphatic Vessel Invasion**	yes	26	53.1
	no	21	42.9
	missing	2	4.1
**Vascular Invasion**	yes	9	18.4
	no	38	77.6
	missing	2	4.1
**First-Line-Treatment**	C	5	10.2
	C+P	15	30.6
	C+P+B	25	51.0
	None	4	8.2
**Relapse after Chemotherapy**	< 6 months	2	4.1
	6–12 months	12	24.5
	>12 months	28	57.1
	none or non-sufficient chemotherapy	7	14.3

n: number of patients, FIGO: International Federation of Gynaecology and Obstetrics, p: pathological, c: clinical, T: extent of primary tumor, N: regional lymph node metastasis, Nx: no evaluation of lymph node status, M: distant metastasis, C: Carboplatin, P: Paclitaxel, B: Bevacizumab.

**Table 2 cancers-11-01250-t002:** Density and spatial distribution of immune cell phenotypes in different lesions of ovarian cancer.

		PT	OM	PE
**Rating**	**CD45 stromal**			
	Mode (Range)	3 (1–5)	3 (2–5)	3;4 (2–5)
	**CD45 intratumoral**			
	Mode (Range)	1 (0–3)	1 (0–3)	1 (0–3)
**Cell Count**	**CD3 stromal**			
	Mean (Range)	626 (5–2491)	1241 (202–4157)	851 (117–2766)
	**CD3 intratumoral**			
	Mean (Range)	201 (0–1134)	212 (0–569)	272 (0–985)
	**CD8 stromal**			
	Mean (Range)	318 (0–1049)	623 (83–1910)	364 (0–843)
	**CD8 intratumoral**			
	Mean (Range)	88 (0–716)	104 (0–636)	130 (0–494)
	**PD-1 stromal**			
	Mean (Range)	73 (0–404)	91 (0–335)	130 (0–601)
	**PD-1 intratumoral**			
	Mean (Range)	26 (0–191)	21 (0–83)	33 (0–145)
**Expression**	**PD-L1**			
	Median (Range)	1% (0–20%)	0.5% (0–20%)	3% (0–20%)

n: number of analyzed tumor samples, PT: Primary tumor, OM: Omental lesion, PE: Peritoneal lesion, PD-1: programmed cell-death protein 1, PD-L1: programmed cell-death ligand 1. To compare primary tumor and corresponding lesions, a Wilcoxon signed-rank test was performed. Results are given in Section 2.3.

**Table 3 cancers-11-01250-t003:** Immune cell phenotypes of primary tumor and corresponding lesions in relation to platinum-sensitivity.

	PT					OM/PT					PE/PT				
			Platinum-Sensitivity *					Platinum-Sensitivity *					Platinum-Sensitivity *		
		n	Red	Full	*p* ^#^		n	Red	Full	*p* ^#^		n	Red	Full	*p* ^#^
**CD45 stromal**		42			1		19			0.350		12			0.250
	**Low**		4	8		**OM ≤ PT**		2	7		**PE ≤ PT**		2	9	
	**High**		10	20		**OM > PT**		5	5		**PE > PT**		1	0	
**CD45 intratumoral**		42			1		19			1		12			1
	**Low**		8	15		**OM ≤ PT**		6	11		**PE ≤ PT**		3	8	
	**High**		6	13		**OM > PT**		1	1		**PE > PT**		0	1	
**CD3 stromal**		42			1		19			1		12			0.523
	**Low**		8	16		**OM ≤ PT**		1	3		**PE ≤ PT**		2	3	
	**High**		6	12		**OM > PT**		6	9		**PE > PT**		1	6	
**CD3 intratumoral**		42			0.057		19			0.633		12			1
	**Low**		11	13		**OM ≤ PT**		2	6		**PE ≤ PT**		2	4	
	**High**		3	15		**OM > PT**		5	6		**PE > PT**		1	5	
**CD8 stromal**		42			0.748		19			1		12			0.523
	**Low**		7	16		**OM ≤ PT**		2	3		**PE ≤ PT**		1	6	
	**High**		7	12		**OM > PT**		5	9		**PE > PT**		2	3	
**CD8 intratumoral**		42			1		19			0.656		12			**0.045**
	**Low**		9	18		**OM ≤ PT**		2	5		**PE ≤ PT**		3	2	
	**High**		5	10		**OM > PT**		5	7		**PE > PT**		0	7	
**PD-1 stromal**		42			0.283		19			1		12			**0.045**
	**Low**		12	19		**OM ≤ PT**		2	4		**PE ≤ PT**		0	7	
	**High**		2	9		**OM > PT**		5	8		**PE > PT**		3	2	
**PD-1 intratumoral**		42			0.738		19			1		12			1
	**Low**		10	18		**OM ≤ PT**		5	8		**PE ≤ PT**		3	7	
	**High**		4	10		**OM > PT**		2	4		**PE > PT**		0	2	
**PD-L1 Positivity**		42			1		19			0.603		12			1
	**No**		5	10		**OM ≤ PT**		5	10		**PE ≤ PT**		2	7	
	**Yes**		9	18		**OM > PT**		2	2		**PE > PT**		1	2	

n: number of patients, red: reduced, PD-1: programmed cell-death protein 1, PD-L1: programmed cell-death ligand 1. * Platinum-sensitivity was defined as follows: reduced (relapse ≤ 12 months after chemotherapy) and full (relapse > 12 months after chemotherapy), ^#^
*p*-value calculated by Fisher’s exact two-tailed test.

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
