# Peer review of "Immune Heterogeneity Between Primary Tumors and Corresponding Metastatic Lesions and Response to Platinum Therapy in Primary Ovarian Cancer"

_cancers, 2019, doi:10.3390/cancers11091250_

Round 1

Reviewer 1 Report

The authors have made all the suggested corrections from original reviewers 1 and 2.  The manuscript would be of high interest in the ovarian cancer field as this type of analysis (immune infiltrate of matched primary and metastatic lesions) had not yet been performed.  The results showing increased infiltrate in metatstatic lesions is particularly interesting as, to date, only lymphocytic infiltrate in primary tumor had been correlated with outcome however the results showing higher PD1 in metastatic infiltrate will have, as the authors suggest, impact in the post hoc analysis of many of these checkpoint inhibitor trials which have been initiated in ovarian cancer.  As a suggestion to the authors for future studies, it would be wonderful to obtain these tissues after the trials had been completed.

Reviewer 2 Report

I read the response to my previous comments and the revised version of the manuscript and I think that the article can be  accepted in this present form.

Reviewer 3 Report

satisfied with revisions.

This manuscript is a resubmission of an earlier submission. The following is a list of the peer review reports and author responses from that submission.

Round 1

Reviewer 1 Report

Very well written paper that will be well accepted in the ovarian cancer field.

Reviewer 2 Report

This is an interesting paper which analyzes the immune heterogeneity between primary tumors and corresponding metastatic lesions founding a correlation with platinum response.

I have only few comments:

1.       The authors studied the CD45, CD3 and CD8 immune infiltrate. It could be interesting to search for expression of CD4+ leukocyte also and compare it with the other markers. It can be done with a double immunostaining to evaluate different markers in the same slide. This may be important to understand the role of the immune response in the peritoneal and omental tissue. Indeed, as reported in the paper of  Serena-Perez et al 2017, cited by the authors in the discussion section, in these sites  CD4+ regulatory cells may be associated with poor outcome.

2.       In the Materials and Methods section, Immunohistochemistry: If the authors used an automated system for immunohistochemistry, which platform they used? (Ventana, Dako?). This can be critical especially for PD-L1 detection.

Reviewer 3 Report

Study on immune heterogeneity of immune marker expression in 49 ovarian cancer samples, which finds differences between primary and metastatic sites.

The authors may have heard of the paradigm shift in ovarian cancer that is high-grade serous carcinomas arise from the fallopian tube. Hence, the ovarian site is also metastatic. The terminology is outdated.

Instead of citing potentially predatory journals [Ref 1], they should have cited relevant articles in the field e.g  PMID: 12529460, 29049607

Table 1: 90% high-grade serous carcinoma, why mix in "other"

Line 101 sentence starting "primary cancer..." makes no sense

Table 2 does intratumoral means intraepithelial because only intratumoral stroma is considered.

A statistical test should be applied to Table 2.

I do not see the value of Table 3 - low numbers. (red/full is not intuitive)

Line 129 is the main finding "no significant difference in intratumoral counts comparing primary and omential lesions". This has been made before, not cited PMID:21623201

The differences in Figure 2 are not relevant because only intraepithelial CD8+TIL matter PMID: 12529460, 29049607. They have to engage with tumor cells directly.

The authors perform multiple statistical tests without correcting.

Association to X and Y are not relevant regarding survival. Sample size to low to allow for robust outcome association studies.

Adjusted ICC should be reported to make results comparable, see PMID: 27473061

Line 38: the authors conclude that "... might represent a selection marker for personalized therapy." but no idea is given how.

Line 261 "soluble" CD45 is likely non-specific overstaining. Did a pathologist interpreted CD45 and PDL1? It can be challenging to distinguish tumor epithelium PDL1 from macrophages.

How was EPCAM used to identify tumor epithelium?
